# DeepSimplex: Reinforcement Learning of Pivot Rules Improves the Efficiency of Simplex Algorithm in Solving Linear Programming Problems

## Abstract

Linear Programs (LPs) are a fundamental class of optimization problems with a wide variety of applications. Fast algorithms for solving LPs are the workhorse of many combinatorial optimization algorithms, especially those involving integer programming. One popular method to solve LPs is the simplex method which, at each iteration, traverses the surface of the polyhedron of feasible solutions. At each vertex of the polyhedron, one of several heuristics chooses the next neighboring vertex, and these vary in accuracy and computational cost. We use deep value-based reinforcement learning to learn a pivoting strategy that at each iteration chooses between two of the most popular pivot rules – Dantzig and steepest edge. Because the latter is typically more accurate and computationally costly than the former, we assign a higher wall time-based cost to steepest edge iterations than Dantzig iterations. We optimize this weighted cost on a neural net architecture designed for the simplex algorithm. We obtain between 20% to 50% reduction in the gap between weighted iterations of the individual pivoting rules, and the best possible omniscient policies for LP relaxations of randomly generated instances of five-city Traveling Salesman Problem. Our results indicate that learning within combinatorial optimization algorithms is possible and that there is much room for improvement in learning more sophisticated pivoting strategies, especially for larger LP instances.

## 1 Introduction

Machine learning has revolutionized many fields by leveraging large amounts of data to learn functions, replacing approaches which used to rely heavily on hand-designed features. One area where machine learning has not made much of an impact is heuristics used inside combinatorial optimization algorithms such as the simplex or integer programming algorithms.

The No-Free Lunch (NFL) theorem (Wolpert et al., 1997) of search and optimization essentially says there are no general-purpose optimization algorithms that work well on all problems. In practice, this implies that many optimization algorithms rely on several different kinds of hand-designed heuristics. But these lack theoretical guarantees, and they embody a one-size-fits-all approach that cannot specialize to the instance distribution. A clear historical example is that of linear programs (LPs). LPs require specification of a pivoting rule, and decades of research has generated many different options. However, no theory exists for how to choose pivoting rules based on the LP instance distributions encountered in practice, let alone how to design new pivot rules based on properties of the instance family.

We propose to address this issue by using data-driven machine learning approaches to learn such heuristics, based on the data encountered by the combinatorial optimization algorithms in practice. Our approach can be interpreted as instantiating the converse of the NFL theorem: any advances made in quality or speed must be due to the algorithm's specialization to the distribution/family of instances it encounters.

In this vein, here we focus on learning pivot rules for the simplex algorithm for solving LP instances. In particular, we learn new pivoting rule policies that combine existing hand-designed heuristics by training on large data sets of LP relaxations of randomly generated instances of the Traveling Salesman Problem (TSP). Our main contributions are:

- We introduce a data-driven approach to learning pivoting rules that focuses explicitly on maximizing the speed of optimization as measured by a novel wall-time weighted iteration cost.

- We use reinforcement learning to learn a hybrid policy that combines two of the most prominent pivoting rules for the Simplex algorithm - Dantzig and steepest edge. The resultant policy decides when to switch between the two rules based on the LP instance objective value and reduced costs at that time.

- We also employ a novel omniscient oracle-based analysis to gauge just how well our policy does relative to an oracle, testing how difficult it is to learn the ground-truth omniscient policy, and whether there exist learnable patterns in the omniscient policy or whether it is a "random" function that can only be memorized.

- We achieve a 20-50% reduction in the gap between the existing pivoting rules and the best possible (omniscient) policy. To our knowledge, this is one of the first studies to report improvements via learning for combinatorial algorithms.

## 2 RELATED WORK

Algorithms for solving optimization problems, e.g., combinatorial optimization problems, often involve heuristic-based algoritmic strategies where, arguably, it is impossible to find an optimal policy that will improve the performance of the algorithm for a wide range of instances. Machine learning approaches give the possibility of devising data-driven methods for the existing heuristics. Khalil et al. (2016) learn to make branching decisions on the branch-and-bound tree in mixed-integer programming. Bonami et al. (2018) learn a classifier for mixed-integer quadratic programming problems to decide whether linearizing the quadratic objective will improve the performance. Bertsimas & Stellato (2019) solve online mixed-integer optimization problems at very high speed using machine learning. They convert parametric mixed-integer quadratic optimization problems to a multiclass classification problem and obtain two to three orders of magnitude speedups compared to the state-of-the-art solver Gurobi. Bengio et al. (2018) provide a detailed survey of machine learning approaches for combinatorial optimization problems.

TSP is a canonical example of combinatorial optimization problems, where there are recent studies for developing machine learning-based heuristic algorithms (Khalil et al., 2017; Bonami et al., 2018; Hansknecht et al., 2018). Vinyals et al. (2015) develop a supervised learning algorithm for combinatorial problems using "pointer networks" and present computational results for TSPs. Bello et al. (2016) use a reinforcement learning approach where they optimize the parameters of the recurrent neural network using a policy gradient method. They claim that their reinforcement learning approach outperforms the supervised learning algorithm presented by Vinyals et al. (2015).

Concorde TSP solver (Applegate et al., 2006a) is one of the best exact TSP solvers, which uses cutting plane methods (Applegate et al., 2003) on the TSP integer programming formulation, i.e., a lazy implementation of a conventional TSP formulation (details given in Section 3). It iteratively solves linear programming relaxations of the TSP formulation and uses a branch-and-bound tree to reduce the search space for the optimal solution. One of the main motivations of this study is to learn the structure of the LP relaxation of the TSP and speed up solving the iterative LP relaxations by learning a pivoting rule policy.

## 3 LINEAR PROGRAMMING AND TRAVELLING SALESMAN PROBLEM

A general formulation of an LP is as follows:

$$
\begin{aligned}
\text{Minimize} \quad & c^\top x \\
\text{subject to} \quad & Ax = b, \\
& x \geq 0,
\end{aligned}
$$

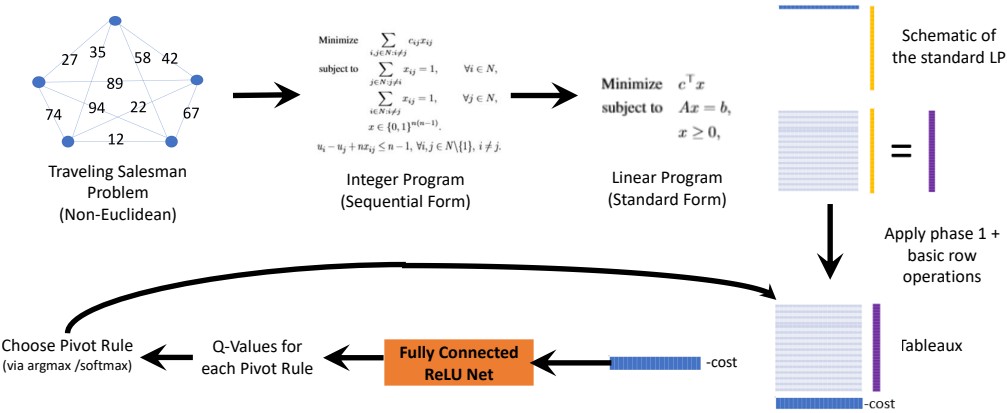

Figure 1: Steps of the Learning algorithm

where the objective $c \in \mathbb{R}^n$, right-hand side $b \in \mathbb{R}^m$, constraint matrix $A \in \mathbb{R}^{m \times n}$, and the number of variables is greater than the number of constraints, i.e., $n > m$. The goal of an LP is to find an optimal decision variable $x$ that minimizes the $c^\top x$ value over the feasibility region $\mathcal{P} = \{x \in \mathbb{R}^n_+ \mid Ax = b\}$. There are different approaches to solve LPs, e.g., the simplex algorithm, interior-point methods, and the ellipsoid algorithm, where the interior-point methods and the ellipsoid algorithm are polynomial algorithms. In this study, we focus on the simplex algorithm, which is the most commonly used method of solving LPs by commercial solvers, e.g., CPLEX and Gurobi.

**Simplex Algorithm.** LPs have attractive properties, including (i) the feasible region is convex, (ii) if there is an optimal solution, there exists an optimal solution that is an extreme point of $\mathcal{P}$, and (iii) an optimal extreme point has at most $m$ non-zero entries, which are called basic variables. The simplex algorithm has two phases. Phase one finds a basic feasible solution, and Phase two finds an optimal basic feasible solution.

The main idea of the simplex algorithm is to find an extreme point and implicitly check its adjacent extreme points. If no adjacent extreme points improve the objective, then the current extreme point is optimal because of the linearity the objective and convexity of $\mathcal{P}$. If there are adjacent extreme points that improve the objective, the simplex algorithm moves to one of them and continues the search until an extreme point with no improving adjacent extreme points can be found. An iteration of the simplex method is given below (Bertsimas & Tsitsiklis, 1997).

1. Form the basis matrix $B \in \mathbb{R}^{m \times m}$ consisting of basic columns of $A$ that are associated with an extreme point (basic feasible solution) $x$.

2. Compute the reduced costs $\bar{c}_j = c_j - c_B B^{-1} A_j$ for all nonbasic indices $j \in \{1, \ldots, n\}$, i.e., $j \in \{1, \ldots, n\}$ such that $x_j = 0$, where $c_B$ is the objective vector associated with the basic entries of $x$ and $A_j$ is the $j^{\text{th}}$ column of $A$. If they are all nonnegative, the current basic feasible solution is optimal, and the algorithm terminates; else, **choose some $j$ for which $\bar{c}_j < 0$**.

3. Compute $u = B^{-1} A_j$. If no component of $u$ is positive, we have $\theta^* = \infty$, the optimal cost is $-\infty$, and the algorithm terminates. If some component of $u$ is positive, let $\theta^* = \min_{\{i \mid u_i > 0\}} \frac{x_{B(i)}}{u_i}$ and $\ell \in \arg \min_{\{i \mid u_i > 0\}} \frac{x_{B(i)}}{u_i}$.

4. Form a new basis by replacing $A_{B(\ell)}$ with $A_j$. If $y$ is the new basic feasible solution, the values of the new basic variables are $y_j = \theta^*$ and $y_{B(i)} = x_{B(i)} - \theta^* u_i, i \neq \ell$.

**Pivoting Rule.** The bold part in Step 2 is the only part of the simplex algorithm that is not precisely specified. The rule of choosing an index is called as the **pivoting rule**. There are many proposed ways to choose the extreme point among the adjacent points in Step 2 above, e.g., Bland's rule, Dantzig's rule, steepest edge rule, and greatest improvement rule. The most commonly used methods for choosing the next extreme point, i.e., the pivoting rule, are Dantzig's rule and the steepest edge rule. Dantzig's rule calculates the rate of improvement (reduced costs) for all nonbasic variables and chooses the best one, i.e., choose $j \in \arg \min \bar{c}_j$, where $\bar{c}_j < 0$, and the steepest edge normalizes the

reduced costs with the norm of their corresponding updated constraint matrix columns and chooses the best one, i.e., choose $j \in \arg\min \frac{\bar{c}_j}{||B^{-1}A_j||}$, where $\bar{c}_j < 0$.

Among the mentioned pivoting rules, only Bland's rule guarantees finiteness; however, it is not practically efficient. For the remaining methods, generally the total number of required iterations of the simplex algorithm decreases as the pivoting rule becomes more complex and requires more time to choose the leaving variable. However, there is no guarantee that a more expensive pivoting rule (in terms of the time it takes to calculate the leaving variable) takes fewer iterations to find an optimal solution. Furthermore, these methods do not guarantee finiteness. Hence, pivoting rules may be improved by learning algorithms.

The simplex algorithm can be viewed as a problem of finding a path on a graph, where extreme points are nodes and only adjacent extreme points are connected. The starting node is the initial extreme point and the end node is an optimal extreme point. The pivoting rules dictate how the next node on the path is selected. Thus, the pivoting rule is a natural candidate for learning algorithms.

**Travelling Salesman Problem.** LPs are essential for solving integer programs (IP) which are LPs for which the decision variables are restricted to take only integer values. Unlike LPs, IPs are $\mathcal{NP}$-hard. The most common way of solving IPs is using branch-and-bound algorithms, which we repeatedly solve linear relaxations of the IP with added inequalities (dropping the integrality requirement and solving the corresponding LP) to obtain feasible integer solutions to form a global lower bound and to get local upper bounds with the relaxed problem.

In this study, we focus on the LP relaxation of the TSP. The TSP considers a list of cities on a connected graph and finds the shortest route that visits each city exactly once and returns to the origin city. The TSP is famously hard to solve, which has attracted the attention of many researchers from different fields. The TSP is an IP with many different formulations presented in the literature, e.g., the subtour elimination formulation Dantzig et al. (1954) and the sequential formulation Miller et al. (1960). Both formulations use a set of cities $N = \{1, \ldots, n\}$ where the length of an arc $i, j \in N$ is $c_{ij}$, and define decision variables $x_{ij} = 1$ if and only if $i, j \in N$ is a link in the tour and $x_{ij} = 0$ otherwise. Furthermore, both formulations share the following objective and constraints:

$$\text{Minimize} \quad \sum_{i,j \in N: i \neq j} c_{ij}x_{ij} \tag{1a}$$

$$\text{subject to} \quad \sum_{j \in N: j \neq i} x_{ij} = 1, \qquad\qquad \forall i \in N, \tag{1b}$$

$$\sum_{i \in N: i \neq j} x_{ij} = 1, \qquad\qquad \forall j \in N, \tag{1c}$$

$$x_{ij} \in \{0, 1\}, \qquad\qquad \forall i, j \in N : i \neq j. \tag{1d}$$

The conventional formulation (Dantzig et al., 1954) has the following set of subtour elimination constraints

$$\sum_{i,j \in N: i \neq j} x_{ij} \leq |M| - 1, \ \forall M \subset N\backslash\{1\}, \ |M| \geq 2. \tag{1e}$$

Hence, the conventional formulation has $n(n-1)$ binary variables and $2^n + 2n - 2$ constraints.

The sequential formulation introduces new continuous variables $u_i$ which represents the sequence in which city $i$ is visited for $i \neq 1$. The set of extra constraints of the sequential formulation is

$$u_i - u_j + nx_{ij} \leq n - 1, \ \forall i, j \in N\backslash\{1\}, \ i \neq j. \tag{1f}$$

The sequential formulation has $n(n-1)$ binary and $n-1$ continuous variables, and $n^2 - n + 2$ constraints. The sequential formulation has a polynomial number of constraints compared to the exponential number of constraints of the conventional formulation. However, the conventional formulation is more practical computationally, despite its larger size (Applegate et al., 2006b).

Our study focuses on reducing the solution time of the LP relaxation of the sequential formulation for the TSP. We aim to learn an optimal way of choosing the pivoting rule for the LP relaxation of the sequential formulation which will potentially lead to reduction of the TSP solution time using the sequential formulation.

## 4 LEARNING APPROACH

Our study focuses on reducing the solution time of the LP relaxation of the sequential formulation for the TSP. We aim to learn an optimal way of choosing the pivoting rule for the LP relaxation of the sequential formulation, which will potentially lead to a reduction of the TSP solution time using the sequential formulation.

The main steps of our learning algorithm are illustrated in Figure 1. The first step is generating a TSP instance by determining the number of nodes and assigning distance between them. We formulate the problem using the sequential formulation and take its linear relaxation. We then convert the LP relaxation to the standard form by adding slack variables so that we can use the simplex algorithm to solve the instance. We use the phase one implementation of a linear programming solver to find a basic feasible solution. The focus of this study is learning a pivoting rule for phase two of the simplex algorithm, where the algorithm starts from a basic feasible solution and finds a path to an optimal solution by traveling to a neighboring basic feasible solution in each iteration.

Every basic feasible solution of the LP has its own basis matrix $B$, reduced cost $\bar{c}$, and right-hand side $\bar{b}$. In each iteration in phase two of the simplex algorithm, we pass the reduced cost vector $\bar{c}$ and the objective value to a fully connected ReLU neural network to estimate the Q-Value which decreases as expected weighted distance rises. Based on the Q-Value estimations, we choose a pivoting rule and iterate the simplex algorithm. The algorithm continues to choose a pivoting rule in each step until the simplex algorithm reaches an optimal basic feasible solution.

## 5 EXPERIMENT DESIGN

**Training Data:** We generate TSP instances with five cities. We use five-city TSP instances because they are the largest TSP instances where we can build and store the exact graph of the extreme points of the LP relaxation. All instances have fully connected graphs where the cost of traveling between two cities is a random integer value between 1 and 100. We construct 1000 instances where we use 800 of them for training and the remaining 200 for testing.

**Action and State Space:** We consider only Dantzig's and the steepest edge rules because they are the most commonly used pivoting rules in practice. At iteration $t$ of the simplex algorithm, we define action $a_t = 0$ if the Dantzig's rule is selected and $a_t = 1$ if the steepest edge rule is selected. The state space is the set of possible simplex tableaux. We denote the current state at iteration $t$ as $s_t$.

**Choice of metric:** We minimize the total number of weighted simplex iterations. The steepest edge rule is computationally more expensive. Thus we penalize the learner more for choosing the steepest edge rule over Dantzig's rule. We calculate the relative cost of the steepest edge against Dantzig's rule by solving all of the generated TSP instances using purely Dantzig's rule, and purely the steepest edge rule. We calculate the average time of a single iteration for both the Dantzig's rule and the steepest edge rule. In our experiments, we find that the steepest edge rule is 15% more costly than the Dantzig's rule. Hence we weight each steepest edge iteration with 1.15. We note that this relative cost may be hardware and software dependent enabling a user to customize.

**Reward function:** We denote $T$ as the maximum number of unweighted iterations, which is taken as 28 for our experiments to limit the size of Q-values without cutting off a significant portion of paths to optimal solutions, $w$ as the factor by which steepest edge rule is costlier relative to Dantzig's rule, i.e., 0.15, $\ell'(s_t)$ as the objective value before the action is performed, $\ell(s_t, a_t)$ as the objective after the action is performed, $\ell^*$ as the optimal value. Then the reward, denoted as $R(s_t, a_t)$, at iteration $t$ is:

$$R(s_t, a_t) = \begin{cases} 0 & t > T \text{ or } \ell' = \ell^* \\ 1 - \frac{1}{T} & t \leq T, \ a_t = 0, \text{ and } \ell'(s_t) > \ell(s_t, a_t) = \ell^*. \\ -\frac{1}{T} & t \leq T, \ a_t = 0, \text{ and } \ell(s_t, a_t) > \ell^*. \\ 1 - \frac{1+w}{T} & t \leq T, \ a_t = 1, \text{ and } \ell'(s_t) > \ell(s_t, a_t) = \ell^*. \\ -\frac{1+w}{T} & t \leq T, \ a_t = 1, \text{ and } \ell(s_t, a_t) > \ell^*. \end{cases}$$

**Q-Value function:** The true Q-value function, denoted as $Q^*$, is the total of expected discounted future rewards,

$$Q^*(s_{t'}, a_{t'}) = R(s_{t'}, a_{t'}) + \max_{\{a_t\}_{t=t'+1}^{\infty}} \sum_{t=t'+1}^{\infty} \gamma R(s_t, a_t)$$

where $s_{t-1} \to s_t$ is a valid state transition and $\gamma$ is the discount factor. If the optimal policy $\pi^*$'s actions $a^*$ are known in advance, e.g., obtained by using the full simplex graph, then the optimal Q-value function, $Q^*$, can be derived using the following recursion when $\gamma = 1$:

$$Q^*(s_{t+1}, a_{t+1}^*) = Q^*(s_t, a_t^*) + \frac{1}{T} + wa_t^*.$$

**Network architecture:** The network has 8 fully connected hidden layers, where each layer has a width of 128 and ReLU activation functions. The input of the network is the reduced costs and the objective value of at the given simplex algorithm iteration. The width of the output is 2, the predicted Q-Value when choosing Dantzig's rule and the steepest edge rule, respectively. The tanh activation function is applied on the 2 outputs.

## 5.1 LEARNING WITH $Q^*$-VALUE

Unlike usual reinforcement learning (RL) applications, for the linear relaxations of five-city TSP instances, we can generate $Q^*$-values by creating the extreme point graph of each LP instance where edges represent possible transitions using the Dantzig's rule or the steepest edge rule. Since these graphs tell us how many weighted and unweighted iterations are needed to reach the optimal solution from any given current state/tableau/vertex and the action that leads to the optimal solution fastest, we can use these actions to recursively construct the $Q^*$-values. Before attempting to learn the Q-values using RL, we use supervised learning of $Q^*$ to glean insights regarding the "learnability" of $Q^*$ and the improvement potential of RL algorithms for minimizing the total weighted iterations.

**Training Algorithm:** For each LP instance, at each iteration of the simplex algorithm, a random choice of action is taken. The tableaux for that LP are stored and sorted into batches of the chosen batch size. Then the neural network is trained on this data set using supervised learning with $Q^*$-values.

**Loss Function:** For a single epoch of the training algorithm, at each iteration of simplex, the neural network generates estimated Q-Values for the simplex tableau. The action corresponding to the higher Q-Value is chosen and the error with respect to the $Q^*$-values is noted. The mean square error over all tableaux is reported as the test loss for that epoch.

## 5.2 LEARNING WITH DEEP RL

Learning with $Q^*$-values tests the potential for learning pivoting rules, and learning with Deep RL, without using $Q^*$-values, aims to show whether it is possible to improve the performance of the simplex algorithm using RL.

**Training Algorithm:** We implement a standard Q-learning algorithm with the reward function described above. During an epoch, for each LP, at each iteration of simplex, the tableaux for the LP instances are stored and sorted into batches. Then the network provides estimated Q-Values, where an action is chosen with an epsilon-greedy approach. Q-Values of the next state and therefore the gradients are found using the target network which is updated every 1/2 epoch. Epsilon is decayed appropriately.

**Loss function:** At any $t$ and a tableau $s_t$, we perform an action $a_t$ and transition to tableau $s_{t+1} \leftarrow (s_t, a_t)$. We define the loss as the mean square error between $Q(s_t, a_t)$ and $\max_{a_{t+1}} Q(s_{t+1}, a_{t+1}) - \frac{1}{T} - wa_t$, where $Q$ refers to the network's estimation of $Q^*$.

## 6 EXPERIMENTAL RESULTS

We first give performance results on supervised learning with $Q^*$-values and reinforcement learning algorithms. We then give an overall analysis of the experiments.

## 6.1 Learning with Q\*-values

Figure 2: Performance comparison of the supervised learning with Q\*-values (blue line) against a random pivoting rule (purple line), pure Dantzig's rule (orange line), pure steepest edge rule (green line), and the best possible strategy (red line) for weighted and unweighted iterations on training (first row) and test sets (second row) for 850 epochs.

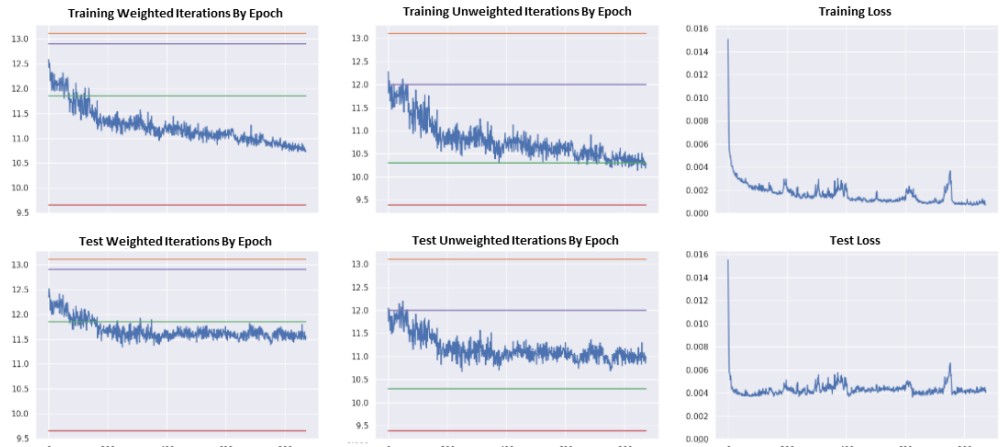

Figure 2 presents the performance comparison of the supervised learning with Q\*-values against pure Dantzig and steepest edge rules, a random strategy where Dantzig's rule or the steepest edge rule is chosen with a 50% chance, and the best possible strategy calculated using the graph representation of the simplex algorithm. The first column of Figure 2 shows the performance of the algorithms for weighted iterations, which is the metric used for training the network. The second column shows the number of unweighted iterations, i.e., the total number of iterations taken. The final column gives the training and test loss in each epoch.

For weighted iterations, the pure Dantzig's rule takes 13.17 iterations, the pure steepest edge takes 11.85 iterations, and the random policy takes 12.9 iterations. The best performing network on the test set takes 11.33 iterations with a corresponding 11.09 iterations on the training set. The network used steepest edge 45.2% of the time on the training set and 47.6% of the time on the test set. The best policy takes 9.65 weighted iterations on average and uses steepest edge 19.1% of the time. Hence, the trained network outperforms all of the pure and random strategies, reducing the difference between the best policy and the pure Dantzig's rule, the pure steepest edge rule and the random policy by 52.3%, 23.6%, and 48.3%, respectively.

The hyperparameters used are a learning rate of 0.0001, a batch size of 128 tableaus, Orthognal Initialisation and Adams Optimizer with default Tensorflow parameters and an L2 regularization of $10^{-7}$.

## 6.2 Deep Reinforcement Learning (RL)

The experiments have the following parameters: 0.001 learning rate, 128 batch size, orthogonal initialization, Adams optimizer and L2 regularisation with $\lambda = 10^{-7}$. We use an epsilon greedy approach to training over time, epsilon begins at 1.00 and is annealed to 0.01 over 50 epochs linearly. The target network is updated twice per epoch.

Figure 3 presents the performance comparison of the deep reinforcement learning algorithm against pure Dantzig and steepest edge rules, the random strategy, and the best policy where their weighted iteration performances on the test set are 13.17, 11.85, 12.9, and 9.65, respectively. The best test performance of the deep RL is 11.25 weighted iterations (using steepest edge 70.1 % of the time) with corresponding 11.17 on the training set (using steepest edge 70.2% of the time), reducing the difference between the best policy and the pure Dantzig and steepest edge rules, and the random policy by 54.5%, 27.3%, and 50.8%, respectively. The last row of Figure 2 shows the RL loss plots on training and test sets. The last column of Figure 2 presents the true loss with respect to the Q\*-values calculated in Section 6.1, indicating that deep RL network successfully estimates Q\*-values.

Figure 3: Performance comparison of the deep reinforcement learning algorithm (blue line) against a random pivoting rule (purple line), pure Dantzig's rule (orange line), pure steepest edge rule(green line), and the best possible strategy (red line) for weighted and unweighted iterations on training (first row) and test (second row) sets for 500 epochs, given with both the reinforcement learning loss and true loss against $Q^*$-values.

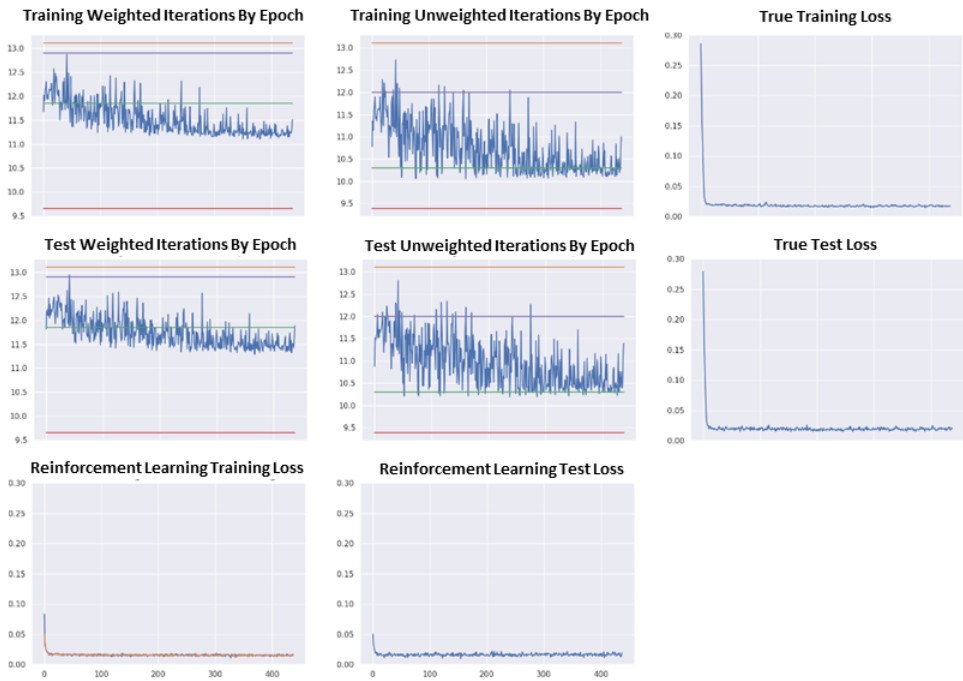

## 6.3 ANALYSIS OF RESULTS

One of the most striking consequences of experiments is that the deep Q network performs on par with the network trained with $Q^*$-values in terms of weighted iterations on the test sets, which suggests that the use of more sophisticated RL frameworks is unlikely to bring significant improvement. The capacity of the network trained using RL to perform as well as supervised learning with $Q^*$-values also suggests scope for success on larger instances, and highlights the need for more bespoke neural network architectures for further improvement.

## 7 CONCLUSION AND FUTURE WORK

We provide the first study on pivoting rules of the simplex algorithm and show the potential for reducing the wall time of solving LPs. Applegate et al. (2003) report that their exact TSP algorithm spends more than 98% of its computational time for solving LPs in large TSP instances. Hence, we believe designing data-driven pivoting rules for a family of LP instances have potential benefits.

In this study, we have focused on small TSP instances (five cities) due to the cost of training neural networks and the difficulty of developing new architectures for this task. Also, omniscient oracle analysis becomes intractable for a larger number of cities. In future work, we want to explore how well our approach scales to larger LP instances and how well it can be incorporated into combinatorial optimization algorithms that solve LPs as a subroutine. The neural network inference has a significant wall time cost, and we do not consider hardware acceleration approaches here. However, the successes seen on mobile devices for image and video recognition networks suggest that if there is something to be gained, then the hardware will follow. Finally, our results rely on the relative wall time estimates, which are, in turn, heavily dependent on the hardware used and the typical instance size in the family. We plan to update our algorithm to learn wall-time costs online.

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
