# OpenReview forum: "DeepSimplex: Reinforcement Learning of Pivot Rules Improves the Efficiency of Simplex Algorithm in Solving Linear Programming Problems"
_ICLR.cc/2020/Conference — Reject_

### Official Review · AnonReviewer2 · 2019-10-23
**Official Blind Review #2**

**Rating:** 1

**Review:**

[ EDIT: Thanks for the response. I still believe the paper is not ready for publication, so I'll keep my rating unchanged. But as I said before, this is a really interesting research direction and I hope the authors will continue this work to collect more results and re-submit it. ]

The paper proposes learning a policy for selecting a pivoting rule to apply at each iteration of the Simplex algorithm for linear programming. Several pivoting rules have been proposed in the optimization literature, and different rules work better than others on different instances. By learning a policy that switches among the existing rules at each step of the Simplex algorithm, it may be possible to construct a pivoting rule that outperforms existing ones. The paper considers learning to switch between the Dantzig rule and the steepest edge rule as an RL problem. Results on 5-city TSP instances show that the learned policy can outperform both rules on a test set with respect to number of iterations.

Pros:
- The problem is very interesting and definitely should be explored further. Learning could potentially help combine rules in an instance distribution-specific manner to construct better pivoting rules.
- The paper made a reasonable attempt to provide sufficient background to understand the problem.

Cons:
- The results on the 5-city synthetic TSP instances are not sufficient. While I understand the motivation for considering small problems to measure the best possible strategy’s performance and to learn on Q* values, I’m not convinced that the insights from such small synthetic problems would necessarily transfer to larger LPs for which the solve time is large enough to be able to afford the overhead of neural network inference to try to reduce it. The learning challenges will likely be very different, and the tradeoff between the inference cost of the neural network and the savings from reducing iterations will also likely be different. Scaling up and neural network inference cost are briefly mentioned in the conclusion, but I believe those are the main challenges.
- Presentation of the results need to be improved significantly. Figures 2 and 3 need to be annotated properly and explained more clearly so that they are easy to understand.

Additional comments:
- Although I’m recommending rejection, the problem is interesting and I hope the authors will continue working on it to develop the ideas further and collect results on larger scale problems.
- For permutation invariance and ability to handle variable-sized inputs, a graph neural network would be a better architecture (see, e.g., Gasse et al., NeurIPS’19).
- It would be helpful to give further details on how the best possible strategy is computed, perhaps as part of an appendix.
- Another baseline to compare against is the performance of an oracle that makes the best possible choice of the pivoting rule per problem instance. This will indicate how well a fixed choice per instance can work if that choice is made as well as possible, compared to switching among choices at each iteration.
- “To our knowledge, this is one of the first studies to report improvements via learning for combinatorial algorithms.” This sentence needs to be clarified because a literal interpretation of it is not true, as shown by, e.g., Bengio et al. https://arxiv.org/abs/1811.06128.

**Experience Assessment:**

I have published one or two papers in this area.

**Review Assessment: Checking Correctness Of Derivations And Theory:**

I assessed the sensibility of the derivations and theory.

**Review Assessment: Checking Correctness Of Experiments:**

I carefully checked the experiments.

**Review Assessment: Thoroughness In Paper Reading:**

I read the paper at least twice and used my best judgement in assessing the paper.

---

> ### Author Response · Authors · 2019-11-15
> **Response to Review #2**
>
> We thank the reviewer for the helpful comments and suggestions.
>
> We added a general response to the scalability concerns.
>
> We are thankful for the graph neural network suggestion. We will implement different architectures in the future to compare performance, including the graph neural network.
>
> We will create a new section that will analyze the learned pivoting rule to gain insights on how the neural network decides one rule over the other.
>
> We will revise our figures to improve their clarification.

---

### Official Review · AnonReviewer1 · 2019-10-25
**Official Blind Review #1**

**Rating:** 1

**Review:**

Summary:
The authors propose a deep-reinforcement learning method for training how to choose pivot rules for the simplex algorithm for a set of LP instances. In particular, the authors applied the RL-approach to randomly generated TSP problems with five cities, which reduces the costs.

Comments:
I have some concerns on the technical contribution.
First of all, I wonder in what kind of realistic situations where we want to “learn” something from optimization problem instances. When we can learn something, the problems would share some properties and there have to be many such instances. I don’ know practical examples of such a scenario. The artificial TSP instances used in the experiments are not convincing for practical applications.

The second concern is the scalability. In the experiments, TSP instances with only five cities are used, for which a simple brute-force search is still acceptable and too small. So, the proposed method is far from practical yet.

As a summary, the idea could be interesting, but the paper is not mature enough.

Comments after Rebuttal:
I read the authors' rebuttal comments. I think the authors' work is valuable and should be investigated further, but the scalablity issues make me feel that the paper is still premature.

**Experience Assessment:**

I have read many papers in this area.

**Review Assessment: Checking Correctness Of Derivations And Theory:**

N/A

**Review Assessment: Checking Correctness Of Experiments:**

I assessed the sensibility of the experiments.

**Review Assessment: Thoroughness In Paper Reading:**

I made a quick assessment of this paper.

---

> ### Author Response · Authors · 2019-11-15
> **Response to Review #1**
>
> We thank the reviewer for the comments.
>
> We added a general response to the scalability questions of the reviewers.

---

### Official Review · AnonReviewer3 · 2019-10-26
**Official Blind Review #3**

**Rating:** 3

**Review:**

The authors present a learned method for speeding up optimization of LP, with an application to the TSP problem. In particular, they discuss choosing a pivoting strategy in the simplex algorithm that is set up as a 2-class classification problem, taking the simplex tableau and outputting one of two classes (Dantzig or steepest edge). They consider both the supervised learning approach where they first compute the optimal policy Q*, as well as the RL approach.

The paper presents a quite interesting approach to solving the TSP, using NNs on top of the tableaus. I enjoyed the presented ideas, however the authors could have done a great job of clearly presenting their work. Notation is not the best, and the experiments are quite limited, indicating limited practical value of the current approach. Detailed comments follow:
- Especially given that the authors have 2 more pages for writing, it would be beneficial to add more visualization and intuitive explanations of the presented ideas.
- The notation should be improved, especially given that there isn't that much notation anyway. E.g., eq (1e) and the following one are not quite clear, what is M? Is it a scalar, or a set? Reading closely it becomes clear it is a set, but the authors use upper-case letter both for scalars and sets, leading to confusion. This is very easy to fix, and would help readers quite a lot.
- The first paragraph in Section 4 is literally a copy-paste of the last paragraph in Section 3.
- Accompanying Section 4 discussion, would be good to add a figure showing an example tableau. Actually, tableau is not even mentioned here and is the main input to the network, it is only mentioned later.
- Using just 5 cities does seem small, indicating limited practical value of the method. Please comment on the actual full size of the exact graph of the relaxation, and why is it infeasible to store. Currently the explanation is hand-wavy.
- "The state space is the set of possible simplex tableau", so you enumerate all the possible tableaus of extreme points prior to training? It seems yes, but would be good to explain that explicitly.
- In "Reward function" section, "as the objective value ...", please add a reference to eq (1a) here. It helps readers follow the text by referencing already introduced parts.
- Where did the equations for rewards come from? They are just given, without much explanation or intuitive discussion.
- Above Section 5.1, "where s_{t-1} -> s_t", where is this notation even used?
- Typo: "value of at"
- Is it "tableau" or "tableaux"? Choose one.
- Section 5.1, are both outputs updated at each iteration? It says that a random action is taken, but is that after Q-values for both actions are checked? It seems yes, but it is not clear.
- In general, a number of small details are missing, and the authors should make sure to not skip them. This introduces confusion to the text which should clearly be avoided.
- "Epsilon is decayed appropriately." How?
- Please define an epoch. In particular, what constitutes a 0.5 epoch, and how does the update actually work here? So you store examples into batches, keep them on the side, and once half an epoch passes only then use them to update the model? It seems yes, but again would be good to be more gentle in explanations.
- Maybe add an algorithm illustrating the methods? In addition to visualization mentioned above.
- Figures are not really super visible. Also, figures should have captions, so text go below the figure.
- Figure 2 is referenced in Section 6.2, where it should be Figure 3.
- "weighted" vs. "unweighted" iteration should be defined earlier. It is nowhere introduced, and while it may be clear what it means, it is not clearly defined and it should be.
- "Adams optimizer"? Do you mean Adam?
- Experiments are somewhat weak, and seem to indicate that the method does not have much practical value. Is the method useful beyond 5-city size?


===== AFTER THE REBUTTAL ======

Thank you for your rebuttal, and especially for the extra experiments! It definitely adds a lot of value to this effort.
The writing could still be improved, as a number of details are missing as explained in the review and to some extent acknowledged by the authors. I liked the idea quite a lot and would not mind seeing it in the conference, but given the number of issues raised by myself and others it seems that the best route forward is rewriting the paper given the inputs by the reviewers and submitting to a future venue.

I will remain at a "borderline" evaluation, and if the other reviewers change their recommendations to (weak) accept I would not mind accepting the paper (assuming all the changes and promised fixes are implemented).

**Experience Assessment:**

I do not know much about this area.

**Review Assessment: Checking Correctness Of Derivations And Theory:**

I assessed the sensibility of the derivations and theory.

**Review Assessment: Checking Correctness Of Experiments:**

I assessed the sensibility of the experiments.

**Review Assessment: Thoroughness In Paper Reading:**

I read the paper at least twice and used my best judgement in assessing the paper.

---

> ### Author Response · Authors · 2019-11-15
> **Response to Review #3**
>
> We thank the reviewer for the valuable comments. The scalability of the results is a common criticism of the reviewers so we added a general response.
>
> Regarding the reward function, the intuition is that at every step there is a small penalty for choosing Dantzig and a penalty that is slightly larger for steepest edge (15% larger since this is the extra cost of SE). There is also a large reward of +1 if the optimal solution is found. We will clarify this in the paper.
>
> Since for larger size instances there is no good estimate of how long instances could take, one would have to have a termination condition (when after some finite number of steps the optimal solution isn’t found. If after this many steps, the optimal solution isn’t found a reward of -1 is assigned.
>
> To ensure that Q-Values are in [-1,1] we choose the small penalty to be 1 / max_iterations for Dantzig and 1.15 / max_iterations for SE.
>
> Regarding the epsilon decay and epochs, an epoch is one training pass through all 800 LPs in the training set. 0.5 epoch is 400 LPs. The update that occurs every 0.5 epochs is that of the target network, not the network that is actually being trained or for which the errors are estimated. The target network’s parameters are updated with those of the main network every 400 LPs. The main network’s parameters are updated every batch (128 simplex iterations).
>
> “We use an epsilon greedy approach to training over time, epsilon begins at 1.00 and is annealed to 0.01 over 50 epochs linearly.”
>
> Elaborating on this, we know how many iterations are needed to solve the 800 LPs using 50/50 random choice on average. Multiplying this by 50, we get roughly how many simplex iterations occur every 50 epochs. Epsilon is decayed every simplex iteration in a linear fashion so that by this number of iterations, epsilon is 0.01.
>
>
> We use tableau when we refer to a single table, and we use tableaux when we refer to multiple tables.
>
>
> Regarding the simplex tableau, we do enumerate the all set of possible simplex tableaux, which there are more than 250,000 possible realizations. Instead, we focus on simplex tableaux that are reachable by using the Dantzing and the steepest edge rules before reaching an optimal solution. Moreover, the basis and the objective function uniquely identify a simplex tableau, hence we only store these attributes.
>
> Thank you for your comments and suggestions on the figures, we will improve them and add new figures that will help clarify the paper.

---

### Author Response · Authors · 2019-11-15
**Scalability of the Results**

The main criticism we received is that the paper does not give results for larger instances. We would like to explain our motivation to focus on the linear program which is the linear relaxation of five-city TSP modeled by sequential formulation.

The sequential integer programming formulation of a five-city TSP instance has 24 variables, where 20 of them are binary, and 22 constraints. The LP relaxation removes the restriction on the binary variables and instead adds $x_{ij} \leq 1$ for all $i,j \in N : i \neq j$ constraints. Thus, the LP relaxation has 42 constraints. Furthermore, in the simplex representation of an LP, each inequality constraint is assigned with a slack variable. Hence, the LP relaxation of five cities TSP instance under the sequential formulation has 44 variables and 42 constraints in its simplex form. Hence, optimizing pivoting rule choices for the linear relaxation of five-city TSP is not a trivial problem, especially when one considers that there is no intuition for when a given pivoting rule is better than another. Our motivation for choosing 5 city instances is the ability to compare our trained network with the optimal policy which cannot be done for larger TSP instances.

The simplex method is the powerhouse for the branch-and-bound method of solving discrete optimization problems. Even small improvements in its efficiency could have profound effects. This work is an important first step in using deep learning to improve the simplex algorithm.

We believe that our paper gives the right motivation for future research because it (i) presents the first successful attempt at learning a data-based pivoting rule policy, (ii) identifies the gap between the best possible pivoting rules and the existing pivoting rules, (iii) demonstrates the feasibility of learning Q*, and (iv) that reinforcement learning has the potential to be as successful as supervised learning for approximating Q*.

To illustrate our points above we trained the same network with 10 and 15-city TSPs (with the input layer size adjusted for each number) and the same hyperparameters with 2 exceptions:

The maximum number of iterations is 200 for 10 cities and 600 for 15 cities.
The extra cost incurred for Steepest Edge was 58% for 10 cities and 105% for 15 cities.

For 10 city TSPs, the steepest edge needed an average of 39.7 steps (post weighting this becomes 62.7). Dantzig needed an average of 95 steps.

Our learned rule achieved 59.8 average weighted steps on unseen test data.

For 15-city TSPs, Dantzig needed 275 steps on average. SE needed 75 steps which become 153.8 post weighting.

Our learned rule achieved an average of 143 weighted steps on an unseen test set. This implies that our learned rule is still able to obtain modest improvements on larger instances.

We note that these results have been achieved without any tuning for larger instances and that consequently, these results provide a loose lower bound for potential improvement on larger instances.

We thank the referees for their valuable comments on the scalability issue.

---

### Decision · Program_Chairs · 2019-12-19

**Decision:**

Reject

**Comment:**

This paper present a learning method for speeding up of LP, and apply it to the TSP problem.

Reviewers and AC agree that the idea is quite interesting and promising. However, I think the paper is far from being ready to publish in various aspects:

(a) much more editorial efforts are necessary
(b) the TPS application of small scale is not super appealing

Hence, I recommend rejection.